# Spin Reorientation Transition and Negative Magnetoresistance in Ferromagnetic NdCrSb_3_ Single Crystals

**DOI:** 10.3390/ma16041736

**Published:** 2023-02-20

**Authors:** Lei Chen, Weiyao Zhao, Zhaocai Wang, Fang Tang, Yong Fang, Zhuo Zeng, Zhengcai Xia, Zhenxiang Cheng, David L. Cortie, Kirrily C. Rule, Xiaolin Wang, Renkui Zheng

**Affiliations:** 1School of Physics and Materials Science, Guangzhou University, Guangzhou 510006, China; 2Institute for Superconducting and Electronic Materials, Innovation Campus, University of Wollongong, Wollongong, NSW 2500, Australia; 3ARC Centre of Excellence in Future Low-Energy Electronics Technologies, University of Wollongong, Wollongong, NSW 2500, Australia; 4Jiangxi Engineering Laboratory for Advanced Functional Thin Films, School of Materials Science and Engineering, Nanchang University, Nanchang 330031, China; 5Jiangsu Laboratory of Advanced Functional Materials, School of Electronic and Information Engineering, Changshu Institute of Technology, Changshu 215500, China; 6Wuhan National High Magnetic Field Center, School of Physics, Huazhong University of Science and Technology, Wuhan 430074, China; 7Australia’s Nuclear Science and Technology Organization, New Illawarra Rd, Lucas Heights, Sydney, NSW 2234, Australia

**Keywords:** spin reorientation, negative magnetoresistance, Dzyaloshinskii–Moriya (DM) interaction

## Abstract

High-quality NdCrSb_3_ single crystals are grown using a Sn-flux method, for electronic transport and magnetic structure study. Ferromagnetic ordering of the Nd^3+^ and Cr^3+^ magnetic sublattices are observed at different temperatures and along different crystallographic axes. Due to the Dzyaloshinskii–Moriya interaction between the two magnetic sublattices, the Cr moments rotate from the *b* axis to the *a* axis upon cooling, resulting in a spin reorientation (SR) transition. The SR transition is reflected by the temperature-dependent magnetization curves, e.g., the Cr moments rotate from the *b* axis to the *a* axis with cooling from 20 to 9 K, leading to a decrease in the *b*-axis magnetization f and an increase in the *a*-axis magnetization. Our elastic neutron scattering along the *a* axis shows decreasing intensity of magnetic (300) peak upon cooling from 20 K, supporting the SR transition. Although the magnetization of two magnetic sublattices favours different crystallographic axes and shows significant anisotropy in magnetic and transport behaviours, their moments are all aligned to the field direction at sufficiently large fields (30 T). Moreover, the magnetic structure within the SR transition region is relatively fragile, which results in negative magnetoresistance by applying magnetic fields along either *a* or *b* axis. The metallic NdCrSb_3_ single crystal with two ferromagnetic sublattices is an ideal system to study the magnetic interactions, as well as their influences on the electronic transport properties.

## 1. Introduction

The *RMX*_3_ (*R* = rare-earth elements, *M* = Fe, Cr, V, *X* = O, Sb) compounds possess two magnetic sublattices, whose moments are ordered at different temperatures, and always show temperature and magnetic field dependent mutual interactions. Benefiting from the dual magnetic structures, physics phenomena such as Dicke cooperativity effect [1], ultra-strong magnon–magnon coupling [2], and spin-phonon coupling [3] have been observed in *R*FeO_3_ single crystals. Between the two magnetic sublattices, one of the most important interactions is the antisymmetric exchange interaction, namely the Dzyaloshinskii–Moriya (DM) interaction, which contributes to the spin reorientation (SR) transition in *RMX*_3_ compounds [4,5,6,7,8]. The antiferromagnetic structure in YFeO_3_ also yields polarized spin waves diffusing over distance and, therefore, can be used in spintronic applications [9].

Another interesting compound family with dual magnetic sublattice is *R*CrSb_3_, which crystallizes in an orthorhombic structure with space group *Pbcm*, and possesses a quasi-2D crystal structure with the infinite *R*Sb and CrSb_2_ layers stacking along the *a* axis [10]. The two magnetic sublattices order ferromagnetically at different temperatures. The moments of the Cr sublattice are ordered along the [010] direction below ~100 K while the moments of the rare-earth *R* sublattice are ordered along the [100] direction below 20 K [10]. Interestingly, below the magnetic ordering temperatures of *R* ions, the Cr moments also become ordered ferromagnetically along the [100] direction [10,11]. Due to the interplay between Cr 3*d* electrons and rare-earth 4*f* electrons, *R*CrSb_3_ single crystals show a series of interesting magnetic phenomena. As of now, the *R*CrSb_3_ family with *R* = La, Ce, Sm, Gd, Pr etc. have been well-studied. The LaCrSb_3_ compound, the first member of this family has been conjectured to have itinerant electrons and local moments of Cr. The latter order ferromagnetically and antiferromagnetically at 125 K and 96 K along the *c* axis, respectively [10]. Recent studies on the *R*CrSb_3_ family reveal an interesting interplay between the moments of Cr and those of rare-earth elements, which induces attractive electronic transport and magnetic behaviors [12]. 

Previously, Deakin et al. reported the magnetic and magneto transport properties of the NdCrSb_3_ crystal, and neutron diffraction was also used to analyze its magnetic structures [13]. Their results demonstrated that the Cr moments show ferromagnetic order at 107.8 K with the easy axis of magnetization along the *b* axis, and that the Nd moments order ferromagnetically at 12.7 K along the *a* axis. They also revealed that the spins of Cr^3+^ couple with those of Nd^3+^ as the temperature is cooled down, and finally results in the change of the easy axis of magnetization from the *b* axis to the *a* axis. Later, Inamdar et al. [10] fabricated high-quality single crystals of NdCrSb_3_ via self-flux method and investigated the anisotropic behaviors of the magnetic susceptibility *χ*(T), isothermal magnetization *M*(H), electrical resistivity *ρ*(T), and heat capacity *C*(T). They concluded similar ferromagnetic transition temperatures of Cr and Nd moments as those obtained by Deakin et al., and attributed the complex magnetic phase diagram of the NdCrSb_3_ crystal to the interplay between the moments of Cr and those of Nd.

In this work, we reported the growth of high-quality NdCrSb_3_ single crystals and their electronic transport and magnetic properties. Employing pulsed high magnetic fields and neutron scattering, we revisited the magnetic structures of NdCrSb_3_, and focused on the SR transitions with temperature and magnetic field. We observed negative magnetoresistance in SR region (~9–20 K), reflecting relatively disordered spin configuration within this region. The observed interplay between Cr and Nd moments and their coupling to the transport properties may inspire more study of this family of ferromagnetic metallic compounds.

## 2. Experimental Details

Single crystals of NdCrSb_3_ were synthesized by a Sn-flux method. High-purity Nd, Cr, and Sb powder with a molar ratio of 1:1:3, together with Sn flux were carefully mixed and loaded into an alumina crucible in a glovebox. The crucible was placed into a quartz tube which was subsequently sealed under a high vacuum atmosphere. The quartz tube was heated to 1273 K in 12 h, kept at this temperature for 24 h to allow sufficient homogenization, and then cooled down to 923 K in 350 h, at which the residual flux was separated from the crystal surfaces using a centrifuge (Shanghai Precision Instruments S-50 D, Shanghai, China). The temperature dependence of the magnetization and resistivity along different crystallographic axes were measured using a physical properties measurement system (Quantum Design PPMS-14, San Diego, CA, USA). The magnetic hysteresis loops along the *a*, *b* and *c* axis were measured using a vibration sample magnetometer (Quantum Design VSM, USA) equipped on PPMS. The temperature-dependent elastic neutron scattering was measured using a triple axis neutron spectrometer (Taipan [14] and Sika [15]) in the Australian Nuclear Science and Technology Organization (ANSTO). The high magnetic field magnetization was measured at Wuhan National High Magnetic Field Center using a pulsed high magnetic field facility (Wuhan, Chian).

## 3. Results and Discussion

Figure 1a schematically illustrates the alignments of the Cr^3+^ and Nd^3+^ spins at different temperatures. The low-temperature ferromagnetic state is demonstrated in the left panel of Figure 1a, in which the Nd^3+^ moments are ordered along the *a* axis, and Cr^3+^ moments are also forced to align along the *a* axis by Nd^3+^ moments (<9 K). Due to the ordering of Nd^3+^ moments, the ground-state spin configuration should be coaligned, namely, the Cr^3+^ moments are dragged from the *b* axis to the *a* axis by the Nd^3+^ ions with larger moments, As cooling from ~20 K, the spontaneously ordered Nd^3+^ moments increase fast, and are comparable to the moments of Cr^3+^, leading to the SR transition in the 9–20 K region due to the DM interaction. Note that the Nd^3+^ ions possess both spontaneous moments below *T*_C_ ~ 12 K and induced moments by the ferromagnetically ordered Cr^3+^ moments in the 9–20 K region. Upon cooling from 20 K, the Nd^3+^ moments along the *a* axis increase (Figure 1c), not only enhancing the DM interaction but also forcing Cr^3+^ moments rotate from the *b* axis to the *a* axis. Above 20 K, the Nd^3+^ moments are induced by the ferromagnetically ordered Cr^3+^ moments. These spin configurations are decided by the temperature-dependent magnetization study, and will be discussed in Figure 1c in detail. Figure 1b shows the temperature-dependence of the intensity of the neutron diffraction (300) peak of the crystal. Upon cooling, the intensity increases below 100 K and reaches a saturation value at ~10 K, indicating an ordering of the magnetic moments in the *a* plane. Note that the structural (300) peak is allowed; however, its intensity is very weak. Therefore, the increase in the intensity of (300) peak below 100 K is mainly due to the magnetic phase transition. The region indicated by the purple color in Figure 1b corresponds to the SR transition below, in which the moments align parallel to the *a* axis, leading to a decrease in the neutron diffraction intensity of the (300) peak. 

Figure 1c shows the anisotropic magnetization of a NdCrSb_3_ single crystal. It can be seen that the NdCrSb_3_ single crystal is easy magnetized along the *a* axis below ~10 K due to the ordering of Nd^3+^ moments. The decrease of the *b-*axis magnetization indicates that the Cr^3+^ moments are forced to rotate from the *b* axis to the *a* axis, thereby contributing to the ferromagnetic behaviors along the *a* axis together with Nd^3+^ moments. Upon heating, the spin reorientation transition occurs in the 9–20 K temperature range, where the Cr^3+^ moments rotate back to *b* axis from *a* axis. After that, the Cr^3+^ moments dominate the magnetization behaviors resulting in the *b* axis being the easy magnetization axis, and a small part of Nd^3+^ moments are induced by Cr^3+^ moments in the same direction. This can be demonstrated by the field-cooled (FC) and zero-field-cooled (ZFC) magnetization curve along the *b* axis. As shown in Figure 1c, the magnetization reaches ~2 $\mu_{B}/\;f.u.$ at 20 K for the FC mode. This value is more than two times higher than that for the ZFC mode (~0.8 $\mu_{B}/\;f.u.$). When the crystal was cooled in a magnetic field of 0.05 T, the Nd^3+^ moments rotate to parallel with the Cr^3+^ moments, resulting in higher magnetization values. While the crystal was cooled in zero magnetic field, the Nd^3+^ moments distribute spontaneously antiparallel with Cr^3+^ moments, leading to lower magnetization. This phenomenon was previously observed in DyFeO_3_ orthoferrites [4], confirming that the Nd^3+^ moments distribute spontaneously antiparallel with Cr^3+^ moments, corresponding to the smaller ZFC magnetization, and can be easily reversed by an applied magnetic field, resulting in the two-times larger FC magnetization.

Figure 1d shows the temperature dependence of the resistivity, with the electric current along the *a* and *b* axis, respectively. Upon cooling, the resistivity shows appreciable drops in the spin reorientation temperature near 100 K and 12 K, owning to the reduction of the carrier scattering induced by the spin ordering of Cr^3+^ and Nd^3+^, respectively. Note that the resistivity does not show obvious anisotropy.

Due to the different preference of moments, the field-induced SR transition is also expected. Therefore, we further measured the M-H loops along three crystallographic directions at base temperature (2 K) with the magnetic field applied up to 30 T, and the results are shown in Figure 2. The M-H loop along the *a* axis is a typical ferromagnetism, owning to the ordering of both Cr^3+^ and Nd^3+^ moments along this direction. Upon sweeping the magnetic fields, the magnetization reaches a saturation value of ~4.2 $\mu_{B}/\;f.u.$ along the *a* axis, which is, as discussed above, contributed by both Nd^3+^ and Cr^3+^ moments. With further increasing of the magnetic field, the magnetization along the *a* axis increase slightly to ~5 $\mu_{B}/\;f.u.$, which agree with that reported in the literature (~4.2 *μ_B_* at saturation point) [9,13]. Interestingly, the M-H loop along the *b* axis shows different behaviors. The Cr^3+^ moments are believed to be firstly dragged to *b* axis by small magnetic fields, showing a rapid increase in the magnetization. Then a component of Nd^3+^ moments along the *b* axis is induced by the magnetic field up to 1.2 T, after which the magnetization increased dramatically until ~2.5 T. The magnetization along the *b* axis keeps going up slowly with increasing magnetic field, meaning that there are magnetic moments constantly emerging in this direction which may belong to the Nd^3+^. As the hard magnetization axis, the *c*-direction M-H loop along the *c* axis shows paramagnetic behavior. Note that there is an unusual increasing of magnetization between 10 T and 13 T, which is probably due to the spin reorientation of Cr^3+^ moments. The magnetization value at *B* = 30 T reaches the similar value of about 5 $\mu_{B}/\;f.u.$ for all the three crystallographic directions, which could be the total magnetic moments in the NdCrSb_3_ single crystals.

Figure 3 shows the magnetic hysteresis loops at different temperatures and the magnetoresistance and Hall resistivity, which were measured by applying the magnetic field along the *a* and *b* axes, respectively. As shown in Figure 3a, the hysteresis loops along the *a* axis at temperatures below 20 K are square-like, revealing presence of ferromagnetism in this direction, agreeing with the high magnetization value along the *a* axis in Figure 1c. The *a*-direction becomes paramagnetic after the crystal was heated to temperatures above the spin reorientation temperature (~20 K), and this is because the ordering of the Nd^3+^ moments disappear above ~20 K and the Cr^3+^ moments return to the *b* axis. Figure 3b shows the M-H loops along the *b* axis. Although the magnetic moments parallel to the *a* axis at temperatures below 20 K, the M-H loops along the *b* axis still show the ferromagnetic feature, and the reason to explain this is that the Cr^3+^ moments can be easily dragged back to the *b* axis by applied magnetic fields. With the temperature rising, the direction of all the Cr^3+^ moments switch to the *b* axis, making the *b* axis being the easy magnetization axis until about 120 K where the ordering of Cr^3+^ moments disappear. Note that the components of Nd^3+^ moments, which are induced by Cr^3+^ moments, also contribute to the magnetization in Figure 3b.

Figure 3c,d show magnetoresistance (MR) along the *a* and *b* axes at different temperatures, respectively. There are negative MR values within the spin reorientation temperature region (around 10~20 K) due to the fragile magnetic structure at this temperature region. As aforementioned, the Cr^3+^ and Nd^3+^ moments are comparable in this temperature region, which results in the flexible magnetic structure. Without applied fields, the moments are relatively disorder, and contributes to large electron scattering. The negative MR along the *a* axis is approximately 10% at 10 K with 2 T, while MR along the *b*-direction is about 5%. The negative MR has also been reported in Cr_2_Se_3_ [16] and Ti-doped Cr_2_Se_3_ [17] crystals due to the similar reasons. MR beyond the spin reorientation temperature range for both two directions are positive, further supports the high disorder states in SR region. The Hall effect along the two directions is shown in Figure 3e,f, from which anomalous Hall effects are observed. In the RFeO_3_ system [9], the spintronic application requires heterostructure building due to the insulating nature of the crystals. In metallic NdCrSb_3_ crystal, the intrinsic negative MR and anomalous Hall effect directly reflects the spin information, and may potentially benefit the spintronic applications.

## 4. Conclusions

In this work, we fabricated high-quality NdCrSb_3_ single crystals and measured their magnetic properties and electronic transport properties. Neutron diffraction is used to detect the temperature-dependent intensity of the (300) peak. The behaviors of Nd^3+^ and Cr^3+^ moments are investigated in detail. We found out that below 20 K, the Nd^3+^ moments order along the *a* axis, and the Cr^3+^ moments are also forced to this direction by the Nd^3+^ moments. The magnetic moments experience reorientation at the temperature range of 9~20 K, where the moments can be easily aligned along the applied field direction. Upon heating, the Cr^3+^ moments turn to *b* axis from *a* axis at the spin reorientation temperature, and a component of Nd^3+^ moments are induced in this direction. The Nd^3+^ moments are spontaneously antiparallel to the Cr^3+^ moments, but can be easily reversed by magnetic field. The fragile magnetic structure in SR region leads to a negative MR effect and anomalous Hall effect, which may indicate the potential spintronic application in metallic dual magnetic lattice systems.

## Figures and Tables

**Figure 1 materials-16-01736-f001:**
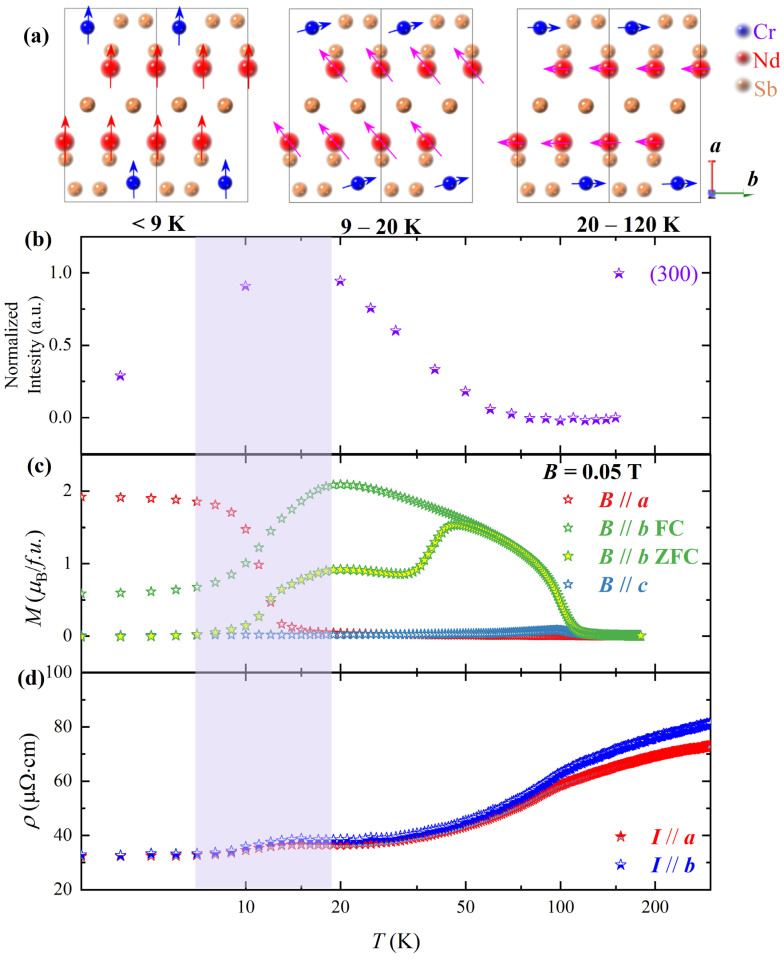
(**a**) The magnetic configuration showing the arrangement of both Nd^3+^ and Cr^3+^ moments along the *a* axis at temperature below the ordering temperature of Nd^3+^ ions. With heating, spins of Cr^3+^ switch from the *a* axis to the *b* axis at the spin reorientation temperature region, and spins of Cr^3+^ totally arrange along the *b* axis with a small amount of Nd^3+^ antiparallel with Cr^3+^ above 20 K. Note that, the pink arrows indicate the Nd^3+^ moments fully or partially induced by Cr^3+^. (**b**) Temperature dependence of the neutron diffraction intensity of the magnetic (300) peak. (**c**) Temperature-dependent magnetization along three directions of the NdCrSb_3_ single crystals with a 0.05 T magnetic field applied. The ZFC and FC curves along *a* and *c* axes coincide, respectively, while those along *b* axis show obvious difference. (**d**) Temperature dependence of the resistivity with the electric current applied along the *a* and *b* axes, respectively.

**Figure 2 materials-16-01736-f002:**
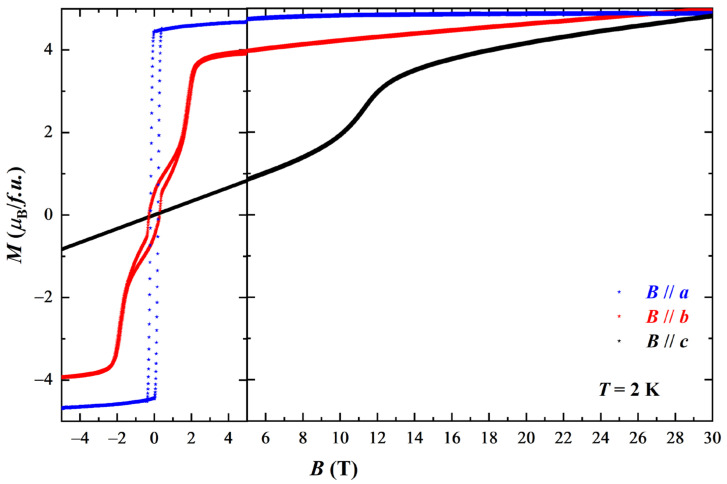
Magnetization as function of magnetic field along three directions at 2 K. left panel (±5 T): magnetization measured by PPMS; right panel (5–30 T): magnetization measured by a pulsed high magnetic field facility.

**Figure 3 materials-16-01736-f003:**
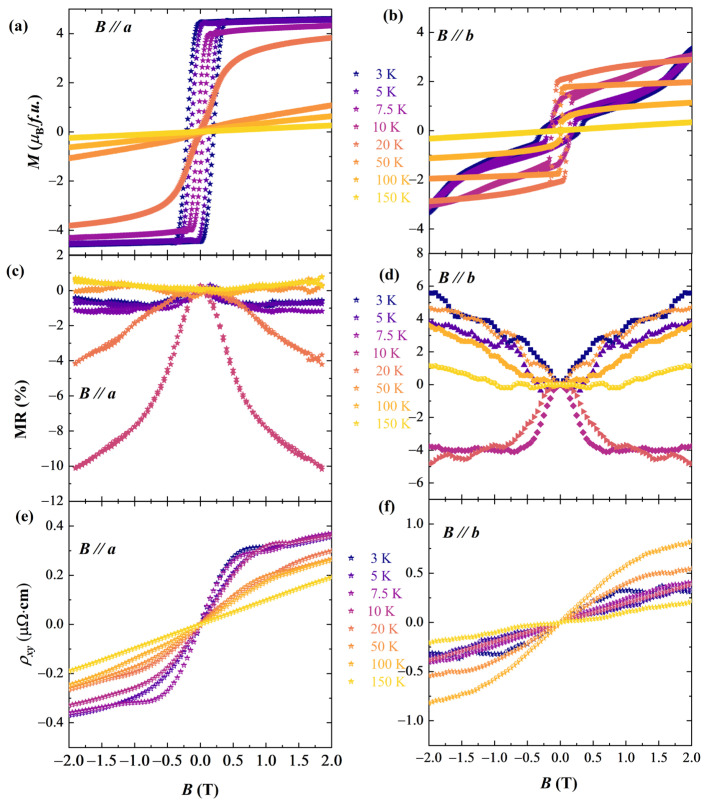
(**a**,**b**) Magnetic hysteresis loops, (**c**,**d**) magnetoresistance, and (**e**,**f**) resistivity of the NdCrSb_3_ single crystals with applied magnetic fields along the *a* and *b* axes, respectively.

## Data Availability

The data are available upon The data that support the findings of this study are available from the corresponding author upon reasonable request.

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
