# Peer review of "Spin Reorientation Transition and Negative Magnetoresistance in Ferromagnetic NdCrSb3 Single Crystals"

_materials, 2023, doi:10.3390/ma16041736_

Round 1

Reviewer 1 Report

In this manuscript, the authors present that they fabricated high-quality NdCrSb3 single crystals and measured their magnetic properties as well as electrical transport properties. 

I recommend this manuscript can be published in Materials after minor revisions.  Please check following comments and questions;

1) In caption of Fig. 1, there is no sentence for (b).  (B)->(c), (c)->(d).

2) In discussion for Fig. 1, the authors state that “drop down drastically at 20K and 100K”.  I cannot recognize such drastically drop down. 

3) In Fig.3(d); MR at B//b, why the MR is large to positive 6% at 3K?

4) In the discussion for Fig.3 (e) and (f), how we can see the anomalous Hall effect from the figure. Please clarify the phenomena.

Author Response

we thank the reviewer for the good suggestions to improve the manuscript.

Reviewer 2 Report

The search for emergent materials with dual magnetic sublattices for potential spintronic applications is of great interest both from the fundamental point of view of magnetic properties and their effect on electronic transport. The manuscript reports the preparation and physical characterization (magnetic properties and electronic transport) of NbCrSb3 single crystals by means of electron neutron scattering, magnetization and resistivity, including experiments at high magnetic fields. The main finding of this work is that the magnetic moments of both Cr3+ and Nd3+ can be reoriented in the presence of a magnetic field leading to high intrinsic negative magnetoresistance and anomalous Hall effect. The paper is well written and reports interesting and novel results in the context of spintronics that are relevant for the target audience of the journal.

Therefore, I am pleased to recommend it for publication in Materials journal, just after the authors take into account the minor points listed below.

-        P3 Replace “The schematics” by “the scheme”

-        Check the superscripts in Nd3+ and Cr3+, some of them are not in the correct format.

-        P4 “The resistivity does not show obvious anisotropy”, why? This observation must be explained.

-        FIGURE 1 caption (a) “from left to right are spins of both”. This is not well formulated. Please use “spin orientation” or “magnetic configuration showing the alignment of spins”

-        FIGURE 1 caption (c) There is a typo, should be “current” instead of “currant”

-        P.6. Replace “literatures” by “literature”. Please add the value reported in the literature to facilitate the comparison.  

-        FIGURE 2 caption “three crystallographic directions” instead of “three directions”

Author Response

We thank the reviewer for the good suggestions.
